# Extraction of Valuable Biomolecules from the Microalga *Haematococcus pluvialis* Assisted by Electrotechnologies

**DOI:** 10.3390/molecules28052089

**Published:** 2023-02-23

**Authors:** Adila Gherabli, Nabil Grimi, Julien Lemaire, Eugène Vorobiev, Nikolai Lebovka

**Affiliations:** 1Université de technologie de Compiègne, UTC/ESCOM, TIMR (Transformations Intégrées de la Matière Renouvelable), 60200 Compiègne, France; 2CentraleSupélec, Laboratoire de Génie des Procédés et Matériaux, Centre Européen de Biotechnologie et de Bioéconomie (CEBB), Université Paris-Saclay, 3 Rue des Rouges Terres, 51110 Pomacle, France; 3Laboratory of Physical Chemistry of Disperse Minerals, F. D. Ovcharenko Institute of Biocolloidal Chemistry, NAS of Ukraine, 03142 Kyiv, Ukraine

**Keywords:** microalga, *Haematococcus pluvialis*, extraction, astaxanthin, biomolecular composition, electrotreatment

## Abstract

The freshwater microalga *Haematococcus pluvialis* is well known as the cell factory for natural astaxanthin, which composes up to 4–7% of its total dry weight. The bioaccumulation of astaxanthin in *H. pluvialis* cysts seems to be a very complex process that depends on different stress conditions during its cultivation. The red cysts of *H. pluvialis* develop thick and rigid cell walls under stress growing conditions. Thus, the biomolecule extraction requires general cell disruption technologies to reach a high recovery rate. This short review provides an analysis of the different steps in *H. pluvialis’s* up and downstream processing including cultivation and harvesting of biomass, cell disruption, extraction and purification techniques. Useful information on the structure of *H. pluvialis’s* cells, biomolecular composition and properties and the bioactivity of astaxanthin is collected. Special emphasis is given to the recent progress in application of different electrotechnologies during the growth stages and for assistance of the recovery of different biomolecules from *H. pluvialis*.

## 1. Introduction

In recent years, the up and downstream processing problem of *H. pluvialis* microalgae has attracted great attention [1,2,3,4,5]. The low-cost bio-production of natural astaxanthin using this microalga is still one of the market’s most concerning issues. It remains a challenge to biosynthesize natural astaxanthin using microalgae such as *H. pluvialis* on a large scale. The main reason for the high cost of natural astaxanthin is that the recovery of the final product necessitates additional manufacturing steps.

The cultivation of *H. pluvialis* for natural astaxanthin production (known as a king of antioxidants) involves two phases, green and red. During the second one, the red cysts or aplanospores accumulate a high amount of astaxanthin, which accounts for ≈80% of the total carotenoid content. Accumulation of astaxanthin is a rather complex process that typically includes specific abiotic stress applications such as nutrient starvation [6], pH [7], illumination stresses [8,9,10], temperature [11], irradiation or pressure stresses [12] and a mixture of different stresses [13]. Particularly, the synergetic combination of different factors allowed improving of astaxanthin accumulation [13,14].

Application of electrotreatments during the growth stages allows genetic transformation, inactivation of culture contaminants, improvement of growth kinetics, accumulation of bioactives and increasing of permeability of cell walls [15,16,17]. The cells of *H. pluvialis* are covered by extraordinarily thick, rigid and indigestible cell walls, and the availability of internal biomolecules is very limited [18,19]. Therefore, the selective recovery of these biomolecules, and particularly astaxanthin, from *H. pluvialis* requires application of rather complex cell disruption and purification techniques. The topic is a very hot issue in contemporary studies of extraction from various microalgae [19]. The processing of *H. pluvialis* using pulsed electrotechnologies (pulsed electric fields and (or) high-voltage electric discharges) is very attractive because with proper adaptation these techniques allow effective extraction and obtaining of high purity extracts [20]. However, the thick cell walls of *H. pluvialis* cysts can restrict the extraction efficiency and applications of pulsed electrotechnologies. Therefore, careful adaptation and optimization of such treatments is required [21]. Nowadays, the existing literature on the application of pulsed electrotechnologies for *H. pluvialis* is very limited.

In this review, the analysis of different steps of up and downstream processing in *H. pluvialis* biorefinery including cultivation and harvesting of biomass, cell disruption, extraction and purification techniques is presented. The useful information on the structure of *H. pluvialis* cells, biomolecular composition, properties and bioactivity of astaxanthin is collected. Special emphasis on the recent progress in the application of different pulsed electrotechnologies for the recovery of biomolecules from *H. pluvialis* is given.

## 2. Main Steps *Haematococcus pluvialis* Biorefinery

At both laboratory and industrial scales, many important steps are included in the processing biorefinery of *H. pluvialis* microalgae. The general overviews of these steps are presented in Figure 1.

Each of these steps may be crucial in determination of the quantity and quality of extracted bioactive molecules. In recent years, the different methods of *H. pluvialis* processing optimization are intensively discussed [1,2,3]. These methods include optimization of favorable and unfavorable growing conditions on cultivation stages and screening of optimized techniques for harvesting of biomass, cell disruption, extraction and purification processes.

## 3. Cultivation and Growth Stages of *Haematococcus pluvialis*, Effects of Electrotreatment

The freshwater microalga *H. pluvialis* was described for the first time in 1841 [22]. It has been reported that this microscopic object can occur in green and red forms. The green forms were observed in juvenile states. In cold seasons (winter, spring, and autumn), the green forms were mainly observed [23,24]. A detailed discussion of the early studies of *H. pluvialis* was presented in 1899 [25]. *H. pluvialis* occurs in different regions of Europe, Africa, and North America [26].

### 3.1. Cultivation and Growth Stages

It is still acknowledged that the green microalga *H. pluvialis* is recognized for its capacity to produce significant quantities of natural astaxanthin via the application of different stress conditions and its high specific growth rate 1.3 d^−1^ [27].

*H. pluvialis* astaxanthin production for the market is relatively new. The induction of astaxanthin bioaccumulation in *H. pluvialis* cells is carried out by subjecting the cells to stress growth conditions. In the last two decades, the two-stage strategy for cultivation has been widely adopted for both laboratory and large-scale cultures. The first stage involves growing *H. pluvialis* under favorable conditions for green biomass production (green stage); this is followed by a second stage (red stage) in which the astaxanthin biosynthesis is induced by changing the favorable growth conditions to unfavorable ones. The natural response to the harsh environmental conditions involves the inhibition of normal growth and the accumulation of astaxanthin for the purpose of protecting the photosynthetic apparatus from photo-oxidative damage by absorbing the excess light. The role of astaxanthin is important to protect cells by scavenging free radicals and reacting with reactive oxygen species. This role is to ensure the maintenance of photosynthesis during red-stage stress [28].

Obviously, the life cycle of *H. pluvialis* microalgae is complicated. The two-stage culturing implies different cell forms. This life cycle can be roughly divided into motile (bi-flagellated cells) and non-motile stages (aplanospores, astaxanthin-accumulating aplanospores, red cysts cells, sporangium and zoospores released from the sporangium) [19,29,30,31]. Each transition from the motile to the non-motile cell form involves remarkable changes in the cell wall structure.

Photosynthetic activity is highest in the initial vegetative green growth phase (days 1 to 12) [7]. In this green vegetative phase, the cells do not accumulate astaxanthin and their carotenoid pattern is mainly composed of lutein (75–80%) and β-carotene (10–20%) [32]. However, the astaxanthin accumulation is triggered in the second phase by specific stress application such as nitrogen starvation, oxidative stress, salinity stress, high temperature and high irradiation. These cited factors significantly affect the astaxanthin accumulation.

The recent literature on cultivation and growth conditions of *H. pluvialis* is rather vast. It includes discussions of the effects of temperature [11], pH [7], nutrient concentration (nitrogen, phosphorus and carbon starvation) [6,33] and illumination conditions [8,9,10,34,35,36]. The effects of light illumination on the growth of biomass and accumulation of bioastaxanthin were analyzed, and it was demonstrated that red light is more effective for increasing the biomass, whereas blue light supports the accumulation of bioastaxanthin [10]. A comprehensive examination of the effects of different photo-, mixo- and heterotrophic cultivation conditions on *H. pluvialis* cell germination has been recently presented [37].

The effects of temperature (20–30.5 °C) on the cell growth and accumulation of astaxanthin has been discussed [11]. It was demonstrated that increased temperature allows enhanced astaxanthin accumulation combined with nitrogen starvation stress. It was demonstrated that the photosynthetic properties are differentially modulated in response to nitrogen starvation/high-light-illumination stress and a correct balance between these stresses is required for efficient astaxanthin production [38]. Effects of nuclear irradiation and a high concentration of carbon dioxide CO_2_ on enhancing the growth rate and astaxanthin yield were demonstrated [12]. Advanced methods for the genetic engineering of *H. pluvialis* were discussed in [39].

The economical two-stage strategy with low light illumination in the initial phase (growth for biomass production) and a combination of high light illumination and elevated carbon dioxide levels (5 or 15%) in the second phase (astaxanthin accumulation) has been tested [8]. The applied procedures showed a significant increase (2–3 times) in the accumulation of astaxanthin. A two-step process was proposed to maximize algal growth and astaxanthin yield [13]. During the first step (biomass production), the different nitrogen sources were tested. During the second step (carotenogenesis induction), a mix of moderate stressors (mild light, nitrogen limitation, and the addition of sodium acetate) was used. The synergetic combination of different factors allowed the promotion of astaxanthin accumulation. The new sequential heterotrophy–dilution–photoinduction cultivation strategy was shown to be rather effective for production of astaxanthin from *H. pluvialis* [14]. The effects of continuous and interrupted (light/dark cycles) illumination conditions on the production of vegetative green cells and astaxanthin accumulation were investigated [40]. It was demonstrated that light/dark cycles are optimal to produce green cells, whereas continuous illumination is better for astaxanthin accumulation. The effects of fulvic acid on biomass growth and astaxanthin accumulation were investigated [41]. It was demonstrated that the astaxanthin content increased by 86.89% in 5 mg/L-treated cells.

Recently, the details of laboratory and commercial scale cultivation procedures in the production of natural astaxanthin derived from *H. pluvialis* have been discussed [42,43,44]. The effects of different cultivation conditions and strategies, technological innovations and types of cultivation systems and an economic assessment for astaxanthin production were also reviewed [45,46,47,48,49,50,51,52].

### 3.2. Electrotreatments during the Growth Stages

The effects of electrotreatments applied during the growth of *H. pluvialis* on cell viability, cell number density and accumulation of astaxanthin attract great attention. A general review of the application of different electrotechnologies for genetic transformation, inactivation of culture contaminants and improvement of growth kinetics of different species of microalgae is given in [15]. Application of electrotreatments of *H. pluvialis* during the different growing stages may reduce the strength and impermeability of cell walls and positively affect the accumulation of bioactives.

The effects of pulsed electric field (PEF) treatments with square wave and exponential decay wave pulses on cell viability during the growth stages were discussed in [53]. Nanosecond PEF treatment (with very short pulses of 25–50 nm and high electric field strength 40 kV/cm) has been applied to *H. pluvialis* cells during the growth stage [16]. It was demonstrated that such treatment can modulate the expression of the astaxanthin biosynthesis genes. PEF treatment increased both the cell mortality (by up to 20%) and astaxanthin accumulation (by 20–30%) from the initial values. Such results were explained by the expression of carotenoid-pathway-related genes.

Electrical treatment of *H. pluvialis* cells was studied in the bio-electrochemical chambers [54,55]. Such treatments enhanced the nitrogen consumption and chlorophyll biosynthesis and resulted in a considerable increase in electro-treated cell number density (by 20%) after cultivation for seven days [54]. However, such treatments did not significantly affect the intracellular astaxanthin content (10% increase). The special adaptation of electrotreatment parameters (electric strength and duration) allowed a significant increase in the astaxanthin content in electrostimulated *H. pluvialis* (by 36.9%) [55].

Application of electrical treatment by low-temperature plasma in *H. pluvialis* biotechnology has been also discussed. It was demonstrated that low-temperature plasma promotes the growth of *H. pluvialis* [56]. The plasma treatment is widely used as a safe and environmentally friendly sterilization technology [57] and may be applied during the growth stages as a strong oxidative stress and mutagenesis tool without damaging the microalgae cells [17]. Atmospheric pressure argon dielectric barrier discharge plasma was used for mutation induction in *H. pluvialis* [58]. The presence of enhanced astaxanthin accumulation in the mutants was observed that was attributed to genetic modification by the mechanism of carotenogenesis. It was indicated that plasma mutation is suitable for effective *H. pluvialis* breeding with promising industrial applications [59]. The plasma treatment effectively improved the growth of *H. pluvialis* and its astaxanthin accumulation; under optimal conditions the astaxanthin content was 51.96% higher than in the starting strain. The possible mechanisms of astaxanthin accumulation induced by plasma treatment were also discussed [60,61].

## 4. Structure of Cell and Biomolecular Composition of *Haematococcus pluvialis*

### 4.1. Cell Structure

Figure 2 gives a schematic presentation of the cell structure of *H. pluvialis* at the resistant cyst stage. On the final aplanospore stage of cyst formation, the rigid and resistant multilayered cell walls are formed [62,63,64,65,66,67]. The total size of the cells at this stage is approximately 20–30 µm [29]. The cell internal compartment contains cytoplasm starch granules, sub-cellular micro-compartments of pyrenoids and astaxanthin deposited in extra-plastidial oil globules (lipid droplets). This internal space is covered by lipid membrane with thickness of ≈5 nm.

The outside part of the cell is covered with an aplanospore cell wall (with total thinness of 2–3 µm). This wall is composed of an outer primary algaenan-based wall (a very stable sporopollenin-like wall), a secondary wall (composed of mannose and cellulose with homogeneous arrangement) and a tertiary wall (composed of mannose and cellulose with heterogeneous arrangement). Therefore, the cell is protected by complex, strong and rigid cell walls.

The three layered cell structure acts as a strong barrier to different external physical and chemical stresses and provides a high resistance against extreme environmental stress conditions [18,19]. Consequently, the cell wall developed is indigestible and reduces the bio-accessibility of the internal biomolecules such as astaxanthin. Moreover, the thickness of the *H. pluvialis* cell wall that is reached after the stress culturing stage is considered as an important bottleneck limiting the large-scale production of natural astaxanthin.

### 4.2. Biomolecular Composition

The chemical constituents of extracellular extracts of *H. pluvialis* were recently analyzed [68]. *H. pluvialis* cell composition includes ash, protein, carbohydrates, dietary fiber, carotenoids and lipids.

Figure 3 presents the relative percentage proportions of ash, protein, carbohydrates total dietary fiber, carotenoids and lipids in the initial biomass and extracts obtained using supercritical CO_2_ extraction at optimized conditions (pressure of 350 bar, temperature of 50 °C, and CO_2_ flow rate of 0.47 Kg/min) for the H. pluvialis red phase [69]. The total dietary fiber and protein were observed as the main intracellular constituents in the initial biomass. Carotenoids include astaxanthin (66–70%), β-carotene (3.5–6.5%) and lutein (27–28%). The content of proteins, lipids and carotenoids in extracts obtained by supercritical CO_2_ was higher in comparison to that of the biomass. An inverse situation was observed for the content of total dietary fibers (58% in the biomass compared with 26% in the extract).

#### 4.2.1. Lipids, Proteins and Carbohydrates

The content of lipids, proteins and carbohydrates in the *H. pluvialis* cells strongly depends on the stage of cell evolution [29,43,70]. Lipid content in the *H. pluvialis* cells may be rather high during the green vegetative stage (≈20–25% of the total biomass). During the red-stage evolution, lipid content increases (≈32–37%) and can be manipulated by operation conditions and the stress imposed. The protein and carbohydrate contents also change with transition from the green phase to the red phase. The protein content is ≈29–45% in the green stage and it reduces up to 17–25% in the red stage. The carbohydrate content is ≈15–17% in the green stage and it increases up to 36–40% in the red stage.

#### 4.2.2. Carotenoids

During the green stage, the other carotenoids in *H. pluvialis* are lutein (≈50–70% of total carotenoids), neoxanthin, β-carotene and violaxanthin (≈8–13% for each of them); there is no astaxanthin content [43,71]. During the red stage, the total carotenoid content is significantly increased (mainly due to the astaxanthin accumulation) and can reach more than 4% of the biomass dry weight. At this stage, astaxanthin is the main cell carotenoid, and the other carotenoids represent ≈5–20% of the total content.

Astaxanthin (the chemical IUPAC name is 3,3′-dihydroxy-β,β-carotene-4,4′-dione) is a liposoluble red dietary carotenoid. It has the molecular formula C_40_H_52_O_4_, a molecular weight of 596.86 g/mol, a boiling point of 774.04 °C, a density of 1.071 g/cm^3^ and a melting point of 182–183 °C. Astaxanthin contains 13 conjugated double bonds with two benzene rings and hydroxyl groups at the ends [72]. Due to the presence of hydroxyl groups, the molecule of astaxanthin is polar. At room temperature (25 °C), astaxanthin is practically insoluble in water, but it is highly soluble in different organic solvents (Table 1).

It was also demonstrated that the solubility of astaxanthin in acidic conditions was 10–20 times higher than in neutral and basic conditions [76].

Astaxanthin is rather unstable and can be degraded during the production process due to heating (even for a very short time), long-term storage, in acidic conditions under ultraviolet irradiation and in the presence of oxygen [77]. The effect of extraction and drying methods on the antioxidant activity of astaxanthin was discussed in [78]. The general overview of different chemical and biochemical properties of astaxanthin is presented in [43].

The microalga *H. pluvialis* can accumulate the optical pure isomer of astaxanthin (3S, 3′S), which has high antioxidant activity (in neutralization of destructive reactive oxygen species) that is approximately 20 times stronger the synthetic astaxanthin [79]. Several studies have shown that astaxanthin accumulated within the cysts of *H. pluvialis* exists in an esterified form, mainly as monoesters (70%), di-esters (25%) and free form (5%) with other occurring carotenoids [48]. The astaxanthin can be esterified on one or both of its hydroxyl groups with various fatty acids such as palmitic, stearic and oleic acids [80].

For the commercialization of natural astaxanthin, the free form is only approved for the market. This means the purification of final astaxanthin product of *H. pluvialis* needs to pass the additional step of de-esterification (known as saponification). Two methods exist for the de-esterification of extracted astaxanthin: de-esterification through enzymolysis using a cholesterol esterase and de-esterification through saponification in darkness at low or room temperature using an alkaline solution of NaOH or KOH in methanol. However, this step of de-esterification could induce a loss of carotenoids due to their degradation. Three factors could negatively affect the final pigments: treatment time, NaOH or KOH concentration and temperature [81].

## 5. Harvesting of Biomass

The harvesting procedure of *H. pluvialis* is a very important step before application of cell disruption technique. It consists of separation of the biomass from the bulk culture medium, which can be achieved using gravity sedimentation, centrifugation, filtration, flotation or flocculation techniques [82,83]. Gravity sedimentation or centrifugation are rather effective for the large cells of *H. pluvialis* and can be successfully employed. The filtration technique is based on the liquid biomass flowing through the membrane. Flotation is a gravity separation technique, and it involves attaching micro-sized air bubbles to microalgae cells. The flocculation technique is based on the aggregation of *H. pluvialis* cells caused by the addition of flocculants. A combination of these techniques can be useful to achieve a good harvest efficiency for *H. pluvialis* [84].

## 6. Cell Disruption, Extraction, Purification Techniques

The industrial perspective of the downstream processing of *H. pluvialis* and different steps for the optimization of bioactive recovery are widely discussed in the current literature [85,86].

### 6.1. Cell Disruption

Cell disruption can use different physical, chemical and biological techniques (Figure 1). Physical treatments including mechanical, thermal and different innovative methods are widely applied for cell disruption and separation of the value-added biocompounds [82,87,88] phycobiliproteins, carotenoids [89] and astaxanthin [19,43,90,91,92,93,94,95]. Traditional mechanical treatments use bead milling, grinding, high-pressure homogenization and hydraulic and screw pressing [96,97]. Generally, mechanical treatments are non-selective, non-toxic and suitable for large-scale production. Mechanical cell disruption requires high energy input. The thermal treatments use conductive, ohmic or radio frequency heating, autoclaving or application of freeze–thaw. These techniques are also non-selective and non-toxic. However, they are highly energy consuming and some of them may cause the thermal degradation of biomolecules. The new innovative methods assisted by electric and magnetic [98] fields, ultrasound [99,100,101] and microwaves [102,103] may be more selective and are non-toxic, but may require more sophisticated equipment and special adaptation for large-scale production.

Chemical treatments use acid, alkalis, detergent, salt, surfactant, nanoparticle, polymer, solvent, ionic liquid, oxidative and osmotic shock treatments [104,105,106,107]. These treatments do not require advanced equipment but are non-selective, toxic, less suitable for large-scale production and may add chemical contaminants to the extracted biomolecules.

Biological treatments use enzymes, viruses and algicidal bacteria [91,108]. They are relatively selective and less toxic compared with chemical techniques, but they are rather expensive, time consuming and less suitable for large-scale production.

All these treatments can be applied individually; however, for enhanced disruption and increased selectivity, combinations of different techniques may be useful.

### 6.2. Extraction

Once the cell walls are disrupted or weakened, the biomolecules with antioxidant and antimicrobial activity, and particularly astaxanthin, must be rapidly recovered from *H. pluvialis* microalgae to prevent their degradation. Popular methods of extraction are subcritical water extraction, supercritical fluid extraction and extraction with solvents and oils. Supercritical carbon dioxide (S-CO_2_) extraction is considered as the most efficient and has been widely used for industrial applications [71,109]. Subcritical water extraction at 200 °C was demonstrated to be a green and fast extraction method to obtain antioxidant and antimicrobial extracts in the red growing phase and permitted obtaining the high extraction yields of vitamin E and simple phenolics (up to 30% of dry weight) [110]. However, astaxanthin destruction is rather significant at high temperatures. In many works, the different modes of S-CO_2_ extraction at moderate critical temperatures and pressures were tested [111,112,113,114,115,116,117,118,119] (see [120] for a recent review). Supercritical CO_2_ is non-polar, non-toxic, not flammable and inexpensive. The solubility of astaxanthin in supercritical CO_2_ is rather high (≈10^−2^ g/L at 25 °C and 100 bar) [121]. In addition, the CO_2_ at a supercritical state has a high diffusion coefficient and a low viscosity that allows its rapid penetration into the porous matrix of *H. pluvialis*. In some works, CO_2_ modified by addition of ethanol [111,114] and ionic liquids [117] was used to enhance extraction efficiency. The economic assessments of the S-CO_2_ extraction of astaxanthin from *H. pluvialis* were analyzed in [119].

In a simple liquid–solvent extraction the biomolecule recovery is commonly assisted by safe solvents (ethanol, ethyl lactate and ethyl acetate) [122,123] with some limitations for the use of more toxic solvents (acetone, toluene, hexane, benzene, petroleum ether, methanol, dimethylaminocyclohexane and methyl-tert-butyl ether) [121].

### 6.3. Purification

The necessity of a subsequent purification step is mainly defined by the selected cell disruption and extraction methods. Typically, the purification methods are based on the application of electrophoresis, membrane separation (ultra- or electro-membrane filtration), ultracentrifugation or liquid multi-phase flotation (for a review, see, for example, [109]).

## 7. Processing of *Haematococcus pluvialis* by Pulsed Electrotechnologies

The modern, innovative cell disruption techniques based on pulsed electrotechnologies (pulsed electric field (PEF), moderate electric field and high-voltage electric discharges (HVED)) have been previously applied to assist the extraction of biomolecules from different types of microalgae [124,125,126,127]. These techniques typically require lower energy input, smaller amounts of solvent and shorter extraction times than traditional methods [20].

PEF treatment is based on the application of short-duration and high-voltage electric pulses. This treatment can cause perforation of the membrane and the formation of nanopores in them (the phenomenon of so-called electroporation). In turn, electroporation facilitates the solvent-mediated extraction of different intracellular biomolecules from the microalgae species. The effects of electroporation significantly depend on the parameters of PEF treatment such as pulse duration *t_i_*, number of pulses *n*, total time of PEF treatment *t*_PEF_ = *nt*_i_, electric field strength *E*, specific energy input *W* and other details of the PEF protocol. The pulses may be unipolar or bipolar and have an exponential or square-wave shape. The pulse duration ranges from several nanoseconds to several milliseconds, and electric strength ranges in the interval *E* = 100 V/cm–80 kV/m with application of moderate (100–200 V/cm), intermediate (typically under 1000 V/cm), high (1–20 kV/cm) and extra-high (above 20 kV/cm) electric fields dependent on the type of treated microalgae. At the moderate specific energy inputs (*W* = 0.5–5 kJ/kg), electroporation may be reversible, and it becomes irreversible at higher specific energy inputs (*W* = 50–200 kJ/kg). PEF-assisted extraction typically involves moderate specific energies below 20 kJ/kg. At the extra-high electric field strength, high-voltage electric discharges typically occur and are accompanied by different phenomena such as pressure shock waves, bubble cavitation, liquid turbulence, etc. HVED treatment can affect the integrity of the cell wall and enhance the extraction of bioactive compounds.

PEF has been frequently applied in previous studies and there are many recent reviews on applications of PEF for treatment of different biomolecules [87,109,128,129], antioxidants [19,130], pigments and carotenoids [131,132,133,134], phycobiliproteins (fluorescent proteins) [89], proteins and carbohydrates [135], polyphenols [136,137] and lipids [138]. Moreover, the efficiency of different electric-field-based techniques for enhanced microalgae biomolecule extractions (PEF, electrolysis, high-voltage electrical discharges and moderate electric fields) were recently compared [139]. In general, PEF-treatment can be considered as a suitable technique for extraction of the intracellular proteins and pigments from microalgae [140].

However, as far as we know, the data reported in the literature on the application of pulsed electrotechnologies for the recovery of biomolecules from *H. pluvialis* concerns just the extraction of proteins [141,142] and astaxanthin [92,143]. It was stated that the larger cells of *H. pluvialis* are more susceptible to electroporation [82]. However, the thick, rigid cell walls of *H. pluvialis* can lead to inefficient extraction of biomolecules even at very high PEF intensities and numbers of pulses [21].

In study of protein extraction, the *H. pluvialis* suspension was PEF treated in distilled water at *E* = 3–6 kV/cm (nine bipolar pulses with duration of 2 ms); this treatment was followed by a 24 h incubation period in salty buffer. Protein recovery was compared for the PEF-treated and untreated samples [141]. PEF treatment significantly increased the protein recovery (by 10–20 times), and the content of extracted proteins significantly depended on the temperature (4 and 20 °C) and the concentration of cells (10^5^ cells/mL or 10^6^ cells/mL). A post-pulse incubation step was found to be necessary to allow oozing. It was assumed that PEF treatment induces some alteration of the cell walls, increasing their porosity to allow leakage of cytoplasmic proteins. In other work, PEF treatment in water at *E* = 1 kV/cm allowed extraction of 46% of the total quantity of proteins (10.2 µg protein per mL of culture) [142]. The non-destructive electroextraction from *H. pluvialis* with cultures recovered after 72 h was observed. A biocompatibility index (fraction of surviving cells after PEF treatment) was estimated.

Rather contradictory results on the PEF-assisted astaxanthin extraction from *H. pluvialis* were obtained [92,143]. For example, a very low astaxanthin extraction efficiency (3.1–6.5%) was observed using PEF pretreatment in methanol (10–30 kV/cm, 30 min) and an additional 2 h incubation [92]. PEF treatment led to the lowest percentage of astaxanthin extractability compared with ultrasounds, high-pressure microfluidization, hydrochloric acid and ionic liquid techniques. For instance, high-pressure microfluidization led to the maximum extractability value of 91.4%. PEF treatment was considered as inefficient for the cell wall disintegration of *H. pluvialis*. In other work, the PEF treatment was applied to extract astaxanthin from the Nordic microalga *H. pluvialis* [143]. A high extraction efficiency (96% of the total carotenoid content) was observed using PEF treatment in water (10 pulses of 5 ms at 1 kV/cm and 20 °C). The value of around of *E* = 1 kV/cm was considered as a threshold for the effective extraction from this Nordic *H. pluvialis* strain. It was demonstrated that PEF treatment resulted in a 1.2-fold more efficient extraction compared with the classical disruption methods (bead-beating and thermal treatments). It was stated that efficient PEF-assisted extraction of carotenoids requires proper optimization of the incubation time after PEF application. The differences between results obtained in [92,143] may reflect the significant variations in the PEF-assisted extraction protocols and differences in the studied strains [19].

## 8. Bioactivity of *Haematococcus pluvialis* Extracts

The bioactivity of biocompounds extracted from *H. pluvialis* was intensively studied in the current literature [69,144,145,146,147,148]. Many of these studies are related with applications of natural bioastaxanthin. Figure 4 demonstrates that the natural astaxanthin has exceptionally high antioxidant activity compared with other known antioxidants. The biosynthetic (natural) astaxanthin produced by *H. pluvialis* has superior therapeutic properties and is highly preferable in comparison with astaxanthin produced by chemical synthesis from petrochemical sources in the form of a mixture of isomers [49]. Moreover, synthetic astaxanthin is not recommended for use as a human nutraceutical supplement [149].

The estimated cost of bioastaxanthin production from *H. pluvialis* is around USD3000–USD3600 kg^−1^ compared with the cost of production of synthetic astaxanthin, which is around USD1000 kg^−1^ [93]. Bioastaxanthin demonstrated significantly stronger (by order of magnitudes) bioactivity than astaxanthin synthesized from petrochemicals (see Figure 4).

The recent advancements in bioastaxanthin production from *H. pluvialis* and the multifunctional applications and world market for it were analyzed in [93,152,153,154]. Natural astaxanthin is more powerful and a stronger antioxidant for the neutralization of destructive reactive oxygen species found in the human body compared with all other antioxidants [26,29,49,77,151]. It is ≈6000 times more active than vitamin C, ≈55 times more active than coenzyme Q10 (CoQ10), 100 times more potent than vitamin E and five times more powerful than β-carotene for the neutralization of destructive reactive oxygen species found in the human body.

Currently, bioastaxanthin is approved by the FDA (United States Food and Drug Administration) as a health food supplement and food colorant for animal and fish feed [155]. Bioastaxanthin has been also approved as a dietary supplement for human consumption in many European countries, the USA and Japan. The recommended or approved doses of astaxanthin range between 2 and 24 mg/day [155].

Bioastaxanthin demonstrates many attractive anti-aging, anti-inflammatory, sun proofing and immune system boosting effects on living organisms. The possibilities and limitations of the use of astaxanthin in food technology were analyzed in [156]. The possible applications of natural astaxanthin also involve the pharmaceutical [144,145,157] and cosmetic [69,147,158] industries. The medical and human health applications [145,159] including anti-cancer, anti-diabetic and anti-atherosclerotic effects and the immune, neuroprotective and genetic engineering potential exerted by astaxanthin have been discussed in [26,145,159,160,161]. The different effects related to the impact of natural astaxanthin on oxidative stresses, aging (skin and brain aging), skin physiology, central nervous system functioning [77] and inflammation-associated diseases [162] were also discussed.

## 9. Conclusions

The freshwater microalga *H. pluvialis* has a large content of natural astaxanthin (bioastaxanthin), which comprises up to 4–7% of its total dry weight. Bioastaxanthin is very powerful and strong antioxidant for the neutralization of destructive reactive oxygen species found in the human body. However, the cells of *H. pluvialis* are covered by thick and rigid cell walls and extraction of intracellular biomolecules is not an easy task. Pulsed electrotechnologies, and particularly pulsed electric field (PEF), are very promising for the extraction of different biomolecules from *H. pluvialis*. However, the thick, rigid cell walls of *H. pluvialis* can effectively protect the cell membrane against electroporation and lead to inefficient extraction of biomolecules even under heavy PEF loads (high intensities and big number of pulses). PEF treatment is usually referred to as a mild treatment method that could not significantly affect the structure of cell walls. The PEF treatment of *H. pluvialis* requires careful optimization of all the PEF treatment parameters, selection of the type of organic solvent and combined application of different cell destruction techniques. Therefore, more careful future studies on the application of PEF in combination with other cell disruption techniques for *H. pluvialis* are needed.

## Figures and Tables

**Figure 1 molecules-28-02089-f001:**
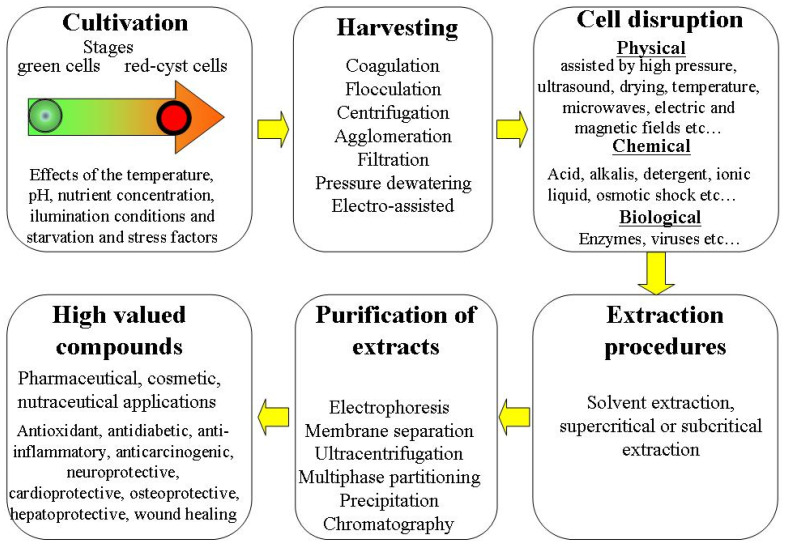
Overview of different steps in *H. pluvialis* microalgae processing biorefinery.

**Figure 2 molecules-28-02089-f002:**
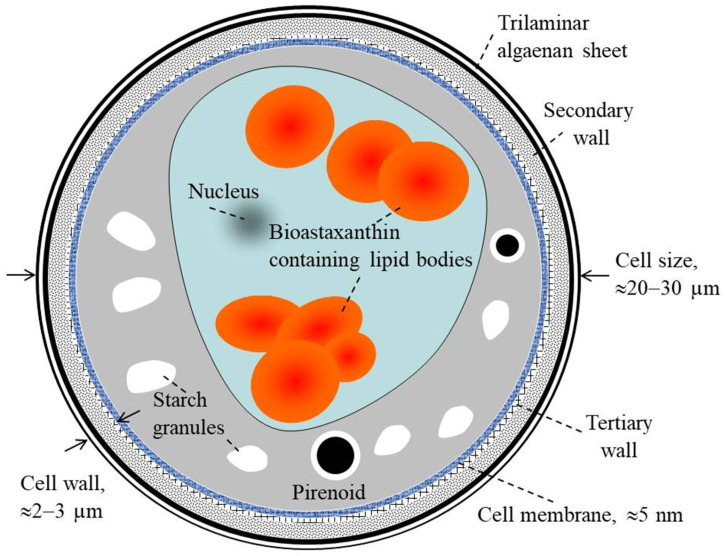
Description of *H. pluvialis* cell structure at the resistant cyst stage. Based on the data collected in [62,63,64,65,66,67].

**Figure 3 molecules-28-02089-f003:**
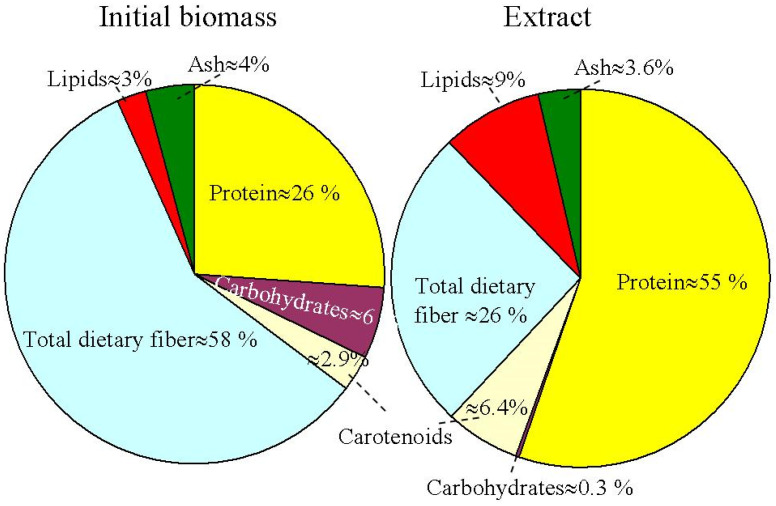
Initial biomass and supercritical CO_2_ extract relative percentage proportions for different biomolecular component compositions in the *H. pluvialis* red phase. Based on the data obtained in [69].

**Figure 4 molecules-28-02089-f004:**
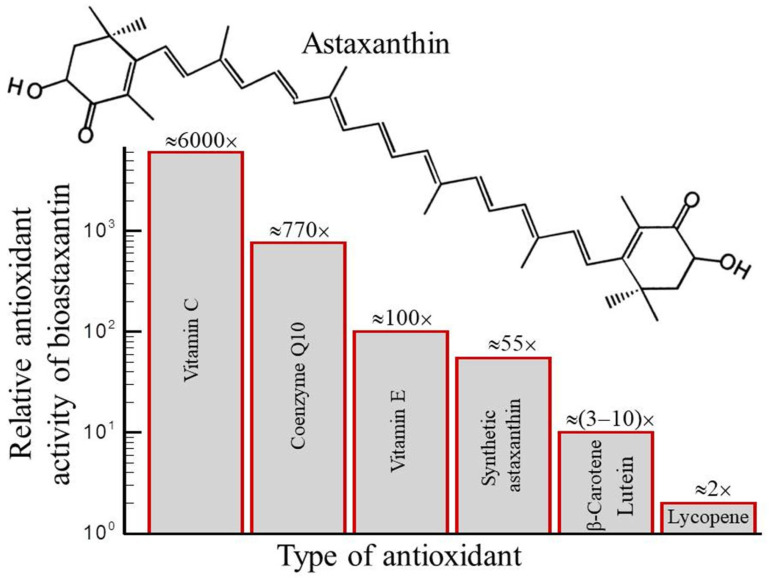
Relative antioxidant activity of natural astaxanthin compared with other antioxidants in the neutralization of destructive reactive oxygen species found in the human body. Based on the data collected in [26,77,150,151].

**Table 1 molecules-28-02089-t001:** Solubility of astaxanthin at room temperature (25 °C) in different organic solvents.

Solvent	Solubility (25 °C), References
Water	≈10^−12^ g/L, [73]
Ethanol	0.038 g/L, [74]
Methanol	0.04 g/L, [74]
Acetone	0.55 g/L, 0.2 g/L, [75]
Dimethyl sulfoxide	1.64 g/L [74], 0.5 g/L [75]
Dimethyl sulfoxide: acetone mixture (1:1)	2.03 g/L, [74]
Chloroform	10 g/L, [75]
Dichloromethane	30 g/L, [75]

## Data Availability

Not applicable.

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
