# Peer review of "Extraction of Valuable Biomolecules from the Microalga *Haematococcus pluvialis* Assisted by Electrotechnologies"

_molecules, 2023, doi:10.3390/molecules28052089_

Round 1
Reviewer 1 Report
Review comments to the author
Title: ''Extraction of valuable biomolecules from microalgae Haematococcus pluvialis assisted by electrotechnologies''.
Manuscript ID: molecules-2219270.
3. Cultivation and growth stages of Haematococcus pluvialis. Effects of electro treatment
1- Page 4, Line 151: Add a dot ''.'' after the citation [45–52].
6.2. Extraction
1- Page 8, Line 326: The temperature '' 25o C'' should be written in right way.
2- Page 8, Line 330: The number ''2'' in the formula ''S-CO2'' should be written in subscript font.
7. Processing of Haematococcus pluvialis by pulsed electrotechnologies
1- Page 10, Line 406: The temperature '' 20o C'' should be written in right way.
8. Bioactivity of Haematococcus pluvialis extracts
1- Page 11, Line 448: The citation (Oliveira et al., 2022) should be written in right way.
Tables:
1- Title of table 1: Add one space before the temperature degree unit ''oC''.
Figures:
- In figure 2, add one space before the species name ''pluvialis''. Please apply this concept in all positions.
- In figure 2, the font is very faint ''re-write the text''.
- In figure 3, the font is very faint ''re-write the text''.
- The resolution and quality of figure 4 should be improved (Re-draw the chemical structure and re-write the text on the left side and under each column).
9. Conclusions:
1- The conclusion should be supported by extra details.
Abbreviations:
- List of abbreviations should be inserted by the end of the manuscript before references.
References
1- All scientific names of species should be written in italic fonts.
2- The scientific names of species should be written in a proper way ''Capitalize the first letter in the genus name and the first letter in the species name should be small'', for example:
- Ref.1: Haematococcus Pluvialis.
- Ref.4: Phaedodactylum Tricornutum.
Author Response
Dear Editor,
Thanks for reviewers’ valuable comments and advices on our paper. All recommendations and specific comments were taken into account and the paper was modified accordingly. We acknowledge the thoughtful suggestions of the Referees, which helped a lot in shaping this paper. See below a detailed list of revisions, changes and comments throughout the text. All changes have been highlighted in yellow colour in the text.
Yours sincerely,
Authors

Reviewer 2 Report
This manuscript "Extraction of valuable biomolecules from microalgae Haematococcus pluvialis assisted by electrotechnologies" reports an interesting review concerning the structure of H. pluvialis, its biomolecular composition, properties, and the activity of astaxanthin. Furthermore, the presentation about electrotechnology and its application throughout the algae growth phases is relevant due to its biotechnological importance and also as a tool to recover bioactive compounds. Overall, the text presents the data adequately. This review has a lot of scientific value, which can still guide the reader regarding the information on the recovery of bioactives from other biomasses.
Author Response

(The authors gave the same response as above.)

Reviewer 3 Report
This manuscript presents a review about the extraction of valuable biomolecules from microalgae Haematococcus pluvialis assisted by electrotechnologies.
This paper provides a review of the analysis of different steps of processing in H. pluvialis biorefinery including cultivation and harvesting of biomass, cell disruption, extraction and purification techniques. The useful information on structure of H. pluvialis cell, biomolecular composition, properties, and bioactivity of astaxanthin is presented. The special emphasis on recent progress in application of different pulsed electrotechnologies for recovery of biomolecules from H.pluvialis was given.
The scientific quality of the manuscript it rises to the scientific level of the Molecules Journal. The technical quality of the manuscript is good in terms of how it was written. The style of expression reflects the scientific training of the authors being in accordance with the requirements of writing the article.
The Abstract is concise and contains sufficient information to highlight the content of the article.
The Introduction provides a clear statement of the problem studied in the present manuscript.
The sections 2-8 are well presented and appropriate for the purpose of research and these follow the guidelines described in the Author Guidelines and they are well presented and discussed.
The authors are advised to carefully check the entire manuscript.
The conclusions of the article are relevant and partially reflect the study conducted.
I also suggest the following:
Line 85, 249: Please check the unit of measure.
The text in figures 2-4 is not visible. Please improve the quality of the figures.
Table 1: Please check if all abbreviations are explained the first time they appear in the text.
Lines 330: Please check the chemical formula.

Author Response

(The authors gave the same response as above.)

Reviewer 4 Report
This review provides concise overview of the production and processing of microalga Haematococcus pluvialis, with emphasis on astaxanthin and eletrotechnologies. It would be of interest to researchers working in the area. The contents are well organized, with good quality of presentation. The few points below would need to be addressed for improving the manuscript:
Figure 1. harvesting box, there are 2 flocculation step - the one after filtration may need to be deleted.
Figure 3. The 2 charts to compare compositions of initial biomass and extract would need to add explanation. I can understand biomass composition, as different methods applied to determine each class of components based on dry biomass. Extracts would be very different situation - you can have aqueous extracts designed to get polysaccharides and proteins out, and not much lipids and carotenoids. On the other hand, you can have non-polar solvent extraction for lipid and carotenoid extraction; this extract will contain no protein or water soluble components. Would the chart on the right side (extract) show the proportion of components (% of dry biomass) obtained or extracted from biomass? Note the content of component in biomass is different from the % of extract obtained.
Line 175-187: level of increase in astaxanthin production with plasma mutation should be given.
Line 226-228: lipids higher in green or red phase? Seems it is higher in red based on the content value provided.
Line 258: 3’S) which is approximately 20 times stronger than the synthetic astaxanthin [79]. Also, stronger in what bioactivity - antioxidant?
Author Response

(The authors gave the same response as above.)
